# A Fresh Look on Bergenin: Vision of Its Novel Drug Delivery Systems and Pharmacological Activities

Sidharth Mehta [1,†] , Varsha Kadian [1,†] , Sweta Dalal [1] , Pooja Dalal [1] , Sunil Kumar [1] , Minakshi Garg [2] and Rekha Rao [1,*]

1 Department of Pharmaceutical Sciences, Guru Jambheshwar University of Science & Technology, Hisar 125001, India; sidharthmehta479@gmail.com (S.M.); kadyanvarsha313@gmail.com (V.K.); swetadalal143@gmail.com (S.D.); lamba808@gmail.com (P.D.); sunilkundu450@gmail.com (S.K.)
2 Department of Pharmaceutics, Delhi Pharmaceutical Sciences and Research University, Ghaziabad 110017, India; minakshigarg@gmail.com
* Correspondence: rekhaline@gmail.com
† These authors contributed equally to this work.

**Abstract:** Bergenin (BER), a key constituent of *Bergenia crassifolia* (Saxifragaceae), has gained extensive attention, owing to its array of pharmacological actions, including anti-infective, anti-cancer, anti-diabetic, neuroprotective, hepatoprotective, anti-urolithiatic, anti-hyperuricemic, and anti-bradykinin properties. Despite ever-intensifying support for its therapeutic features, the poor solubility, lower oral bioavailability, shorter half-life, and more intestinal pH degradation (pH 6.8 or above) of BER have puzzled researchers. To circumvent these pharmaceutical challenges, and to improve its therapeutic efficacy, newer approaches have been adopted by research scientists. Thus, a discussion of the existing literature may provide complete information about the advances in delivery strategies for enhancing its utility. This paper summarizes up-to-date works on the design and development of novel delivery carriers of this bioactive compound, such as phospholipid complexes, extended-release core tablets, prodrugs, herbal gels, polyherbal ointments, nanoparticles, and poly (lactic acid) polymers, with the objective of harnessing its full potential. This review also provides a deep insight into its bioactivities, along with mechanisms. Additionally, the physicochemical attributes, chemistry, and pharmacokinetics of BER are discussed herein. Hence, the comprehensive information documented in this review may introduce new avenues for research advancements of BER.

**Keywords:** *Bergenia crassifolia*; anti-inflammatory; novel carriers; antioxidant; bioactive

## 1. Introduction

With improvements in public awareness, attempts have been continuously made to explore safer alternative remedies for various health issues. Furthermore, the toxic effects of chemically derived drugs and irrepressible risks linked with biological products strengthen the need for an investigation of naturally derived compounds [1]. Nature is a rich source of extremely innovative and diverse bioactive compounds [2]. Plants are incredible in their potential to generate a huge number of specialized metabolites and byproducts with different biological actions. Natural constituents have been used as models for the development of drugs [3] and provided unquestionable support for human welfare [4]. Bergenin (BER) is a natural constituent, which has been extracted from various parts (rhizome, roots, leaves, stem, barks, aerials, seeds, cortex, flowers, wood, tuber, heartwood, fruit or whole plant) of plants [5] such as *Bergenia crassifolia* (*B. crassifolia*), *Bergenia ciliata* (Saxifragaceae), *Corylopsis spicata* (*C. spicata*), *Mallotus philippinensis* (*M. philippinensis*), *Caesalpinia digyna* (*C. digyna*), *Sacoglottis gabonensis* (*S. gabonensis*), and *Mallotus japonicus* (*M. japonicus*) (Table 1 and Figure 1). It is commonly called *Pashaanbheda* (Paashan; rockstone, bheda; piercing) and *Zakham-e-hayat* (zakham; lesion/wound, hayat; life/heal) in the Indian Systems of Medicine [6,7]. It is officially listed in the People's Republic of China (Chinese

Pharmacopoeia Commission, 2010) [8]. In accordance with a citation in the Merck Index, this bioactive compound was firstly isolated from *Saxifraga (Bergenia) siberica* rhizomes [5,9]. BER is trihydroxybenzoic acid glycoside [10]. Traditionally, the rhizomes of *Bergenia* have been used for the treatment of fractured bones, wounds, fresh cuts, pulmonary infections, diarrhea, vomiting, cough, boils, and fever by local people [11,12]. The roots of *Bergenia* have been employed as a deobstruent, demulcent, reliever for ribs and chest pain, a emmenagogue, and a diuretic [6]. The virtues of plants are attributed, to a large extent, to the formation of their secondary metabolites, including bergenin, catechin, and gallic acid, which are mainly used in traditional drugs [13]. BER is a versatile phytoconstituent, as it holds numerous beneficial pharmacological characteristics such as heart disorders, stomach diseases, hemorrhoids, and ophthalmia treatment [2,9]. Additionally, it is accredited with anti-viral, analgesic, antimalarial, antioxidant, and anti-inflammatory potential [7,14]. Owing to these properties, its use as a natural alternative to cure various ailments has increased dramatically in the past decade. Despite the fact that it possesses a wide array of activities, the inherent physicochemical properties of BER limit its pharmaceutical use. The major limitations allied with its delivery are low solubility and poor permeability. Neither highly hydrophilic nor highly lipophilic BER possess poor oral bioavailability. It is commercially available as tablets, pills, and soft gelatin capsules [15] (Table 2, Figure 2), however, the efficacy of these traditional formulations of BER is far lower than expectations [8]. Therefore, novel delivery systems may prove to be promising for overcoming the inherent constraints of BER. It is well known that novel carriers possess a profound potential to improve solubility and stability, modify release behavior, and consequently, enhance the efficacy of entrapped moieties. A handful of BER formulations reported in the literature encompass phospholipid complexes, extended-release core tablets, prodrugs, herbal gels, poly herbal ointment, nanoparticles, and poly (lactic acid) polymers. There is a large number of research and review articles that mainly focus on the role of novel delivery carriers in surpassing the issues associated with bioactive compounds. The available information about BER was collected from popular and widely used databases, such as Web of Science, Google Scholar, Scopus, PubMed, Science Direct, and Springer search. From these searches, a number of citations related to the pharmacological activities, novel formulations, pharmacokinetics, applications, chemistry and physicochemical properties of phytoconstituent BER were retrieved. The keywords used include pharmacological activities, novel carriers, pharmacokinetics, chemistry, patents, and other related words [7]. Herein, we have discussed the outcomes of existing literature to provide comprehensive information with regard to recent advances in BER functionalization, with a specific focus on novel carriers and their potential applications. The aim of this review is to come up with insights into its bioactivities and the mechanism of action. Furthermore, an effort has also been made to touch on the chemistry and pharmacokinetics of BER. Hence, the article will help to identify gaps for future research on this compound.

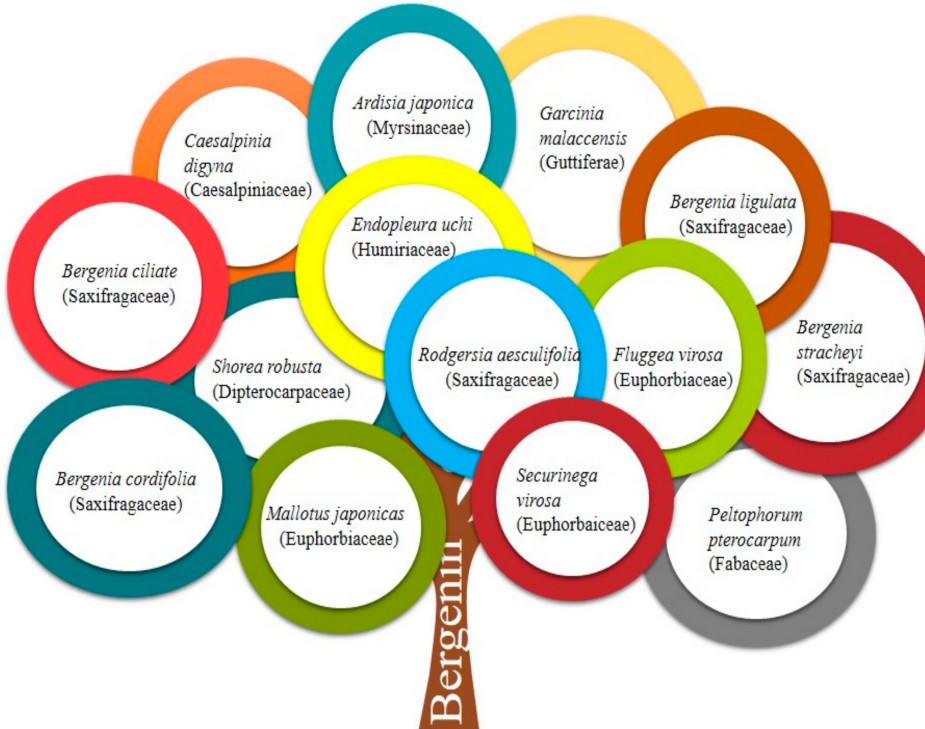

**Figure 1.** Various plant sources of Bergenin.

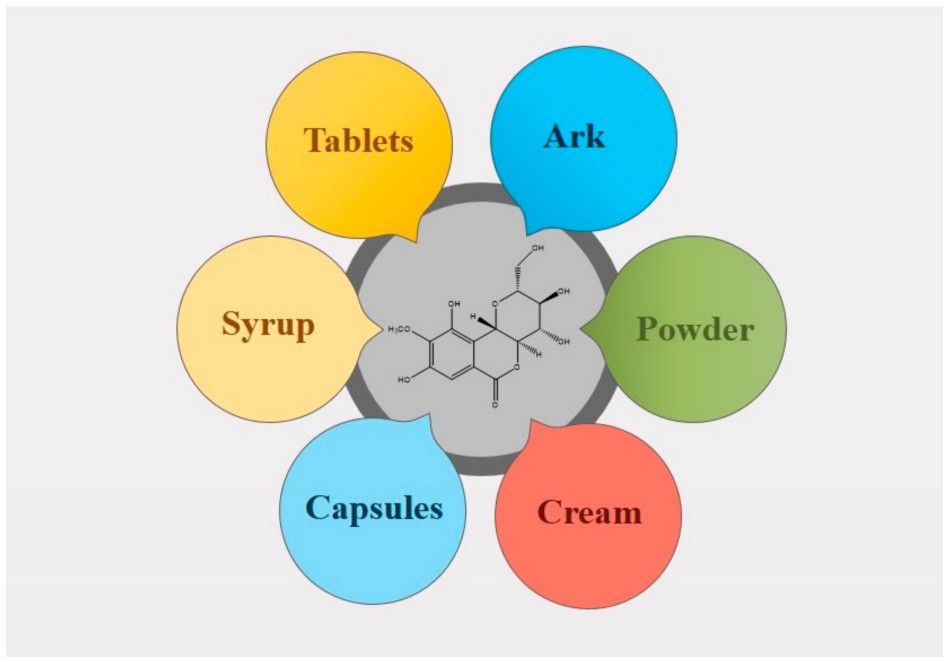

**Figure 2.** Conventional formulations of Bergenin.

**Table 1.** Pharmacological/biological activities of various plants rich in Bergenin.

| Sr. No. | Plants (Families) | Part Used | Pharmacological/ Biological Activities | Mechanism(s) of Action | Study Models | References |
|---|---|---|---|---|---|---|
| 1. | *Bergenia ligulata* | Rhizome | Anti-microbial | Inhibits anaerobic glycosis and aerobic respiration | Agar well-diffusion assay | [16] |
| 2. | *Bergenia* spp. | - | Anti-cancer (cervical cancer) | Inaugural of apoptosis and cell cycle inhibition in the G0/G1 phase. Inhibit phosphorylation of STAT3 proteins. | Cervical cancer cell line HeLa | [17] |
| 3. | *Bergenia ciliata, Bergenia* spp. *Bergenia stracheyi* | Aerial parts - Rhizome | Anti-inflammatory, immunomodulatory | Inhibition of IL-6 and TNF-$\alpha$; Targeting cytokine (IL-1b, IL-6 and TNF-$\alpha$) and reactive oxygen species (ROS), Prevent the development of proinflammatory Th1 cytokines (IFN-$\gamma$, IL-2 and TNF-$\alpha$) whereas potentiate anti-inflammatory Th2 cytokines (IL-5 and IL-4) | Human monocyte leukemia THP-1 cells, Carrageenan-induced paw edema and *Mycobacterium*-induced arthritis in rats; CFA-induced arthritis model | [18–20] |
| 4. | *Bergenia stracheyi, Bergenia ligulata, Bergenia cordifolia, Endopleura uchi, Peltophorum pterocarpum* | Rhizome Bark, Flowers | Antioxidant | Free radical scavenging activity | DPPH assay, Agar well diffusion method, Disc diffusion method | [2,21,22] |
| 5. | *Bergenia cordifolia, Caesalpinia digyna* Rottler | Rhizome Roots | Anti-diabetic | Inhibition of $\alpha$-glucosidase enzyme. Positive effect on endocrine cells of pancreas results in enhanced development of insulin. | Microtitre-based assay. Streptozotocin-nicotinamide induced diabetic rats. | [22,23] |
| 6. | *Mallotus japonicus* | Dried bark | Neuroprotective | Inhibit generation of ROS in brain | Culture of rat cortical neurons in DMEM supplemented with Nitrogen | [24] |
| 7. | *Mallotus japonicas* | Cortex | Hepatoprotective | Attenuated the increase in the activities of alanine aminotransferase, sorbitol dehydrogenase, aspartate aminotransferase, $\gamma$-glutamyltransferase and also inhibit lipid peroxidation and recover the reduced hepatic glutathione level | $CCl_4$-induced hepatic damage in rats | [25] |
| 8. | *Mallotus philippinensis* | Leaf | Anti-urolithiatic | Significantly reduction in calcium, oxalate and phosphate concentration in urine | Ethylene glycol-induced urolithiasis in wistar rats | [26] |

**Table 1.** *Cont.*

| Sr. No. | Plants (Families) | Part Used | Pharmacological/ Biological Activities | Mechanism(s) of Action | Study Models | References |
|---|---|---|---|---|---|---|
| 9. | *Rodgersia aesculifolia* Batal, *Bergenia ligulata* | Rhizome | Anti-malarial | Inhibition of heme polymerization pathway of malaria parasite | In vitro and In vivo assessment of antimalarial activity using *Plasmodium falciparum* and *Plasmodium berghei* infected BALB/c mice | [7,21,27] |
| 10. | *Caesalpinia digyna* Rottler | Root | Anxiolytic | - | EPM (mice) | [7] |
| 11. | *Shorea robusta* | Leaves | Anti-tubercular | Induces the production of TNF-α, NO, IFN-γ, IL-17 and IL-12 from both CD4 and CD8 T-cells | Murine model of *Mycobacterim tuberculosis* infection | [28] |
| 12. | *Bergenia stracheyi* | Rhizome | Anti-gout | Inhibition of xanthine oxidase enzyme | Assayed spectrophotometrically | [2] |
| 13. | *Garcinia malaccensis* | Stembark | Antiplatelet aggregation | Inhibition of platelet aggregation induced by arachidonic acid, adenosine diphosphate and collagen | Platelet aggregation test measured by ANOVA | [3] |
| 14. | *Flueggea microcarpa* | Leaves | Antihyperlipidemic | Reduced level of CH, LDL, VLDL, TG and increased proportion of HDL were reported via, increasing reverse cholesterol transport from arterial tissue to the liver | Albino rats of Charles Foster strain given hyperlipidaemic diet of arachis oil | [29] |
| 15. | *Ardisia japonica* | Aerial parts | Anti-HIV | Inhibition of antibody ADP358 binding to gp120 and interfere with gp120-CD4 interaction | C8166 cells infected with HIV-1 | [30] |
| 16. | *Fluggea virosa* | Aerial parts | Anti-arrhythmic | Coronary artery ligation and blood reperfusion | $BaCl_2$ induced arrhythmia in rats | [31] |
| 17. | *Flueggea microcarpa* | Leaves and roots | Antiulcer | Protection against pylorus-ligated and gastric ulcers induced by aspirin | Gastric ulcers induced by cold restraint stress-induced in guinea pigs and rats. | [32] |
| 18. | *Securinega virosa* | Root bark | Soporific | - | Beam walking test and Diazepam-induced sleeping time assay in mice. | [33] |
| 19. | *Bergenia ciliata* | Rhizome | Anti-tussive | Bronchodilator action, inhibited the histamine and acetylcholine induced contractions | Cough model induced by sulphur dioxide gas in mice | [34,35] |

**Table 2.** Indian conventional formulations for Bergenin.

| Sr. No. | Commercial Herbal Formulations | Name and Amount of Extract Containing Bergenin | Therapeutic Dose Required | Potential Uses/Indications | Manufacturers |
|---------|-------------------------------|------------------------------------------------|---------------------------|----------------------------|---------------|
| 1. | Albestone Capsule | *Pashanbhed (Bergenia ligulata)* 200 mg | As per directed by doctor | Kidney calculi, Bladder calculi, Ureter calculi, Retention of urine, Calculi induced UTI | Sanify Healthcare Pvt. Ltd. (Punjab) |
| 2. | Phytone Capsule | *Pashanbhed Extract (Saxifrage lingulate)* 100 mg | 1–2 Capsules twice daily | Medical Management of Urinary Calculi, for the prevention of recurrent calculi. | Abhinav Healthcare |
| 3. | Stonvil Capsule | *Pashanbhed, (Saxifraga ligulata)* 30 mg | U. T. I.: 2 b. d. for 2 weeks. Renal calculi: 2 b.d. upto 3 weeks. Burning Micturition: 1 b.d. upto 2 weeks. | Burning micturition, Grit, Calculi problems and Urinary tract infections | S.G Phyto Pharma Pvt. Ltd. |
| 4. | Cystone Tablet | *Saxifraga ligulata* (98 mg/tab.) | 2 Tabs. twice daily | Gritty kidney, Ureter, bladder and urethra Sialolithiasi, Urinary tract infection (UTI), Colic ureter, Glomerulonephritis crystalluria—fosfatouria Heart-Renal Edema, Bed wetting-urinary incontinence, Hyperuricemia Enlarged prostate: in concomitant use with speman or Himplasia prevents surgery. | The Himalaya™ Drug Company |
| 5. | Nefrotec~ DS VET Tablet | Pashanbhed (Saxifraga ligulata 30 mg) | Dogs: 1 Tablet two times daily for small breeds. 2 Tablets two times daily for large breeds Cats: 1 Tablet one time a day. | Nephrolithiasis, Recurrent urinary tract infections, Cystitis, Non-specific Urethritis, kidney dysfunction. | The Himalaya™ Drug Company |
| 6. | Neeri Tablet | *Bergenia ligulata* (60 mg/tab.) | Children: 1–2 Tabs. twice a day. Adults: 2–3 Tabs. thrice a day. | Dysuria, Burning Micturition, Crystalluria, Oedema, Anasarca, Non-specific UTIs. | Aimil Pharmaceuticals Ltd. |
| 7. | Patharina Tablet | *Pashanbhed* - | 2 Tablets twice a day orally with water or as directed by the physician. | Kidney Stones, Painful Urination | Shree Baidyanath Ayurved Bhawan Pvt. Ltd. |

**Table 2.** *Cont.*

| Sr. No. | Commercial Herbal Formulations | Name and Amount of Extract Containing Bergenin | Therapeutic Dose Required | Potential Uses/Indications | Manufacturers |
|---|---|---|---|---|---|
| 8. | Cystone Syrup | *Saxifraga ligulata* (53 mg/5 mL) | Children: ½ -1 Teaspoonful (2.5–5 mL) twice daily after meals. Adults: 1–2 Teaspoonful (5–10 mL) twice daily after meals. | Kidney stones, Crystalluria, crystals in the urine, Dysuria, Hyperuricemia, high amount of uric acid in the blood, Burning while urination, Non-specific Urethritis, i.e., irritation or swelling of the urethra. | The Himalaya<sup>TM</sup> Drug Company |
| 9. | Neeri Syrup | *Bergenia ligulata* (100 mg/10 mL) | Children: ½–1 Teaspoonful thrice daily. Adults: 2 teaspoonful thrice daily. | Dysuria, Burning Micturition, Oedema, Anasarca, Non-specific UTIs. | Aimil Pharmaceuticals Ltd. |
| 10. | StonDab Syrup | *Pashanbheda* - | 1–2 Teaspoonful of the syrup 3 times a day. | kidney stones, burning sensation while urination, non-specific urinary tract infection, urinary calculus | Dabar India limted |
| 11. | Ashmarihar kwath Powder | *Saxifraga ligulata* (15 g/100 g) | Mix 5–10 gm of kwath in around 400 mL water and boil it, till residue is 100 mL. | Kidney stone, gall stone problem. | Divya Pharmacy |
| 12. | Pashan Bhed Root Powder | *(Saxifraga ligulata powder roots)* 3 mg | 1–2 Tablespoon mix with water, blend in a smoothie drink/sprinkle over salad. | Urinary tract infection (UTI) burning and painful micturition, spleen related swelling. | Bixa botanical |
| 13. | Pushyanug Churna | *(Saxifraga ligulata)* 5 mg | 2–3 mg, twice a day. | leucorrhoea, menorrhagia, metrorrhagia, prolapse of uterus and also useful in diarrhea, dysentery and bleeding piles | Deep Ayurveda |
| 14. | Prakriti Pashanbhed Ark | *Pashanbhed* - | 10–15 mL of Pashanbhed ark, Twice a day with equal amount of warm water before meals. | Kidney Stones and Liver related problems | Prakriti Nutann Gausadan |
| 15. | Pashan Bhed transdermal Cream | *Pashanbhed* - | Whole spine Swiping downwards 7 times in the morning and evening. | Autoimmune toxins, gall stone, inflammation, kidney stones | Prabhava Ayurvedic herbals |

## 2. Chemical Structure & Physicochemical Properties of BER

BER [IUPAC Name: 3,4,8,10-tetrahydroxy-2-(hydroxymethyl)-9-methoxy 3,4,4a,10b-tetrahydro-2H-pyranol [3,2-c]isochromen-6-one; molar mass: 328.27 gmol$^{-1}$ and molecular formula: $C_{14}H_{16}O_9$] is [15] a white, crystalline powder or loose needle-like crystal powder with a bitter taste and light odor. It becomes discolored upon exposure to heat or light. It has a melting point and a specific optical rotation in the range 232–240 °C and −38° to −45°, respectively [36]. Log *p*-values (−1.060 ± 0.033 to −1.19 ± 0.044) at temperature 37 °C and acidic pH confined the poor lipophilic property of this moiety [37]. It is a C-glucoside of 4-O-methyl gallic acid (2β-D-glucopyranosyl 4-O-methylgallic acid δ lactone) [5]. The initial structures of this molecule were given in 1928 by Tschitschibabin et al. [38] structure I, and Shimokôriyama, structure II, in 1950. These structures were revised by Hay et al. [39], Posternak et al. [40] and Fujise et al. in the year 1959 [41] [Figure 3a]. The conformation of this moiety was unequivocally established by an X-ray analysis of its monohydrate and 3, 4, 8, 10, 11- penta acetate derivatives [42].

(**a**)Structure I (5,7-dihydroxy-6-methoxy -3-(1,2,3,4-tetrahydroxybutyl)-1H isochromen-1-one)

(**b**) Structure II (3,4,5',7'-tetrahydroxy-5 -(hydroxymethyl)-6'-methoxy-4,5-dihydro -3H-spiro[furan-2,3'-isochroman]-1'-one

(**c**) Bergenin

(**d**) Norbergenin

(**e**) Acetylbergenin

**Figure 3.** (**a**,**b**). Initial structures of Bergenin. (**c**–**e**) Structure of Bergenin and their analogs.

The molecule BER is comprised of three six-membered rings: (A) an aromatic ring, (B) an annellated δ-lactone ring, and (C) a glucopyranose ring. The ring (C) is only slightly different to an ideal chair structure. The ring (B) exhibits the predictable half-chair con-

formation. There are six inter- and one intra-molecular hydrogen bonds that outline an extensive hydrogen-bonding arrangement within the crystal. Figure 3b represents the chemical structure of BER and its two analogues—acetylbergenin and norbergenin [43]. Furthermore, Ye et al. have demonstrated that there are six inter- and one intra-molecular hydrogen bond, outlining an extensive hydrogen-bonding net within the crystal, thus it does not have a great number of energetic (active) sites for water sorption [44].

BER possess low aqueous solubility, which results in its poor oral bioavailability. Its poor permeability and low solubility are the major obstacles in its formulation development. Commercially, it is available as soft gelatin capsules, tablets, and pills (Table 2 and Figure 2). Liquid dosage forms for BER are not available on the market, owing to its poor aqueous solubility. The solubility of BER was reported to be highest in polyethylene glycol-400 (PEG-400), followed by dimethyl sulfoxide (DMSO), diethylene glycol monoethyl ether, propylene glycol (PG), ethylene glycol (EG), ethanol, isopropanol (IPA), ethyl acetate (EA), 2-butanol, 1-butanol, and water in a range of temperatures (298.15 to 318.15 K) and pressures 0.1 MPa [15]. Generally, the low water uptake of BER suggested good stability in the presence of moisture during formulation and storage [37].

### 3. Mode of Action of BER

Free radicals are very active molecules that are formed through normal metabolism and cellular respiration. Reactive oxygen species (ROS) are intimately associated with pathological and physiological processes in animals. Chiefly, these species are hydrogen peroxide ($H_2O_2$), hydroxyl free radicals ($OH^-$), superoxide anion free radicals ($O_2^-$), nitrogen oxide radicals ($NO^-$), and others. At lower levels, ROS can work as signaling molecules which control basic cellular mechanisms, such as cellular adaptive and cell growth responses [45]. The excess formation of such free radicals can lead to oxidative injury to biomolecules (proteins, DNA, lipids) [46]. Furthermore, due to imbalances between the body's antioxidant process and the accumulation of ROS, oxidative stress occurs which damages tissues and cells, resulting in the proliferation of numerous ailments. There are clear facts that free radicals are connected with the propagation of ailments, like cancer, atherosclerosis, and emphysema [45].

Bergenin (and its congeners) are extensively employed in Ayurvedic, Traditional Chinese Medicine, Unani, and various folk systems of medicine [5,9]. This bioactive has gained noteworthy attention and is a medicine of choice, by virtue of its multi-target approaches. BER is known for its multiple pharmacological features; anti-inflammatory, anti-oxidative, anti-arthritic, and anti-cancer activities [47], as in Table 1 and Figure 4. In addition, some of the patents related to BER activities are listed in Table 3. The effectiveness of this phytochemical participates in several mechanisms, such as lipid peroxidation inhibitory activity, free radical scavenging activity [6], initiating apoptosis and cell cycle arrest in the G0/G1 phase, inhibiting the phosphorylation of STAT3 proteins, inducing the formation of TNF-$\alpha$, NO, IFN-$\gamma$, IL-17, IL-12, and inhibiting the $\alpha$-glucosidase enzyme (Table 1, Figure 5). All of these have been explained in detail in the sections below [43].

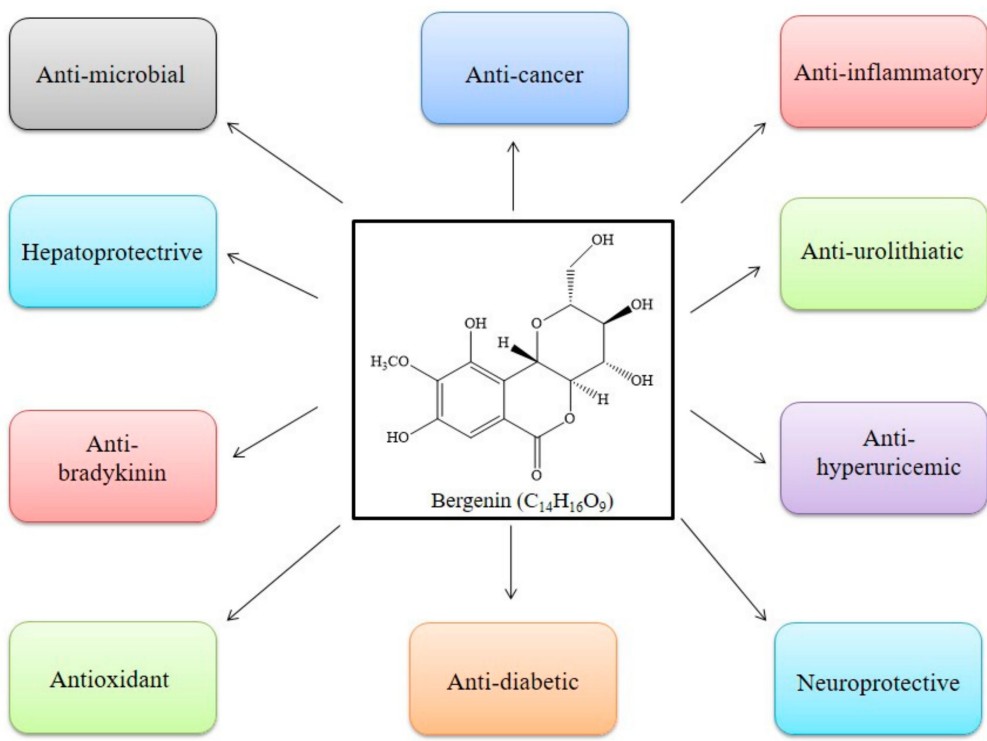

**Figure 4.** Pharmacological activities reported in literature for Bergenin.

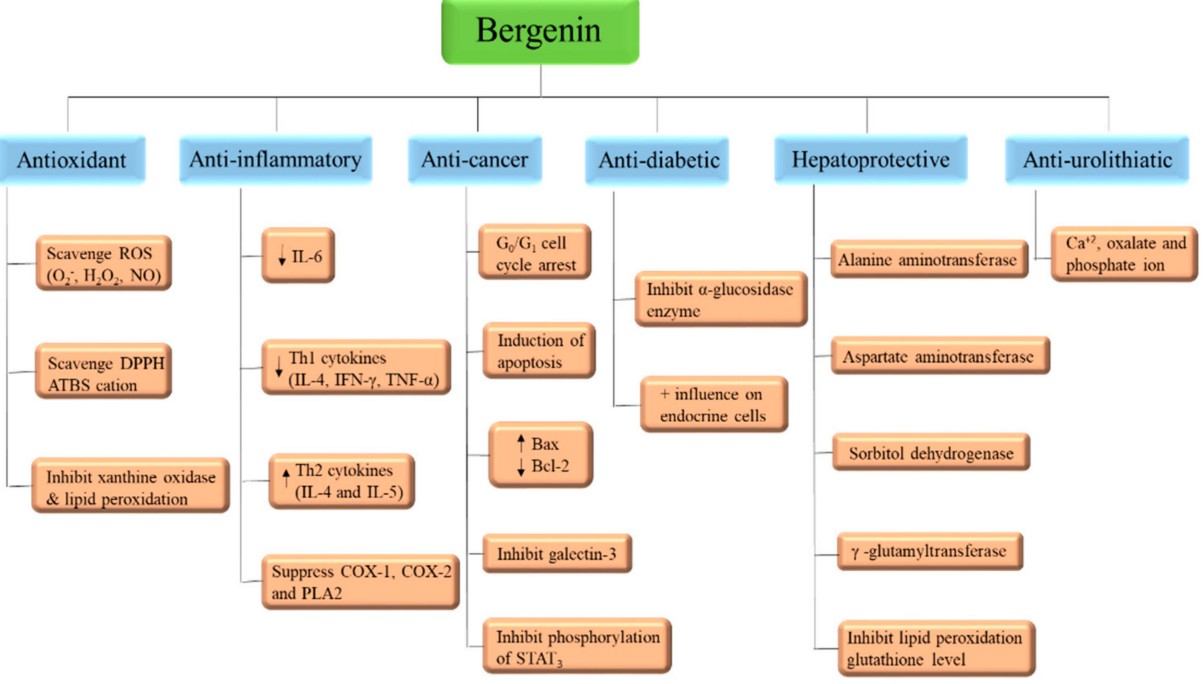

**Figure 5.** Mode of action of Bergenin.

**Table 3.** List of Bergenin Patents.

| Sr. No. | Patents No./Patent Publication No. | Title | Invention | References |
|---------|-----------------------------------|-------|-----------|------------|
| 1. | US 10,494,377 B1 | Bergenin lipoic acid ester with antioxidant activity and a method of preparing the same | Invented BER lipoic acid ester having excellent antioxidant potential. | [48] |
| 2. | US 8,007,837 B2 | Herbal composition for maintaining/caring the skin around the eye, methods of preparing the same and uses thereof | Invented a novel herbal skinceutical composition to maintain and improve skin health especially for delicate skin around the eyes comprising the extracts of *Saxifraga ligulata*, *Cipadessa baccifera* and *Emblica officinalis,* method for preparing the same and use | [49] |
| 3. | US 7,785,637 B2 | Herbal composition for other publications maintaining/caring the skin around the eye, methods of preparing the same and uses thereof | Invented a novel herbal skinceutical composition to improve and maintain skin health especially for delicate skin around the eyes comprising the extracts of *Saxifraga ligulata*, *Cipadessa baccifera* and *Emblica officinalis,* method for preparing the same and their uses. | [50] |
| 4. | 217147 | A Pharmaceutical Composition Useful as an Antioxidant | Invented the process of isolation of BER from *Tinospora crispa.* | [51] |
| 5. | US 2004/O115286A1 | Cosmetic composition of Remedying skin wrinkles comprising bergenia Emeensis extract as active ingredient | Invented a cosmetic composition having Bergenia emeiensisextract, for skin wrinkles owing to its potential inhibition of collagenase and elastase. | [52] |
| 6. | WO2019077620A1 | Gastroretentive sustained release formulations of *Bergenia ciliata* | Invented novel gastroretentive swellable oral formulations for sustained or delayed release of BER-rich *Bergenia ciliata* extract/fraction and a process for preparing the same. The novel formulations were found to be retained in the stomach, which avoids intestinal degradation of BER resulting in its sustained release in stomach over a time period of 16–24 h. | [53] |
| 7. | US 10,494,377 B1 | Bergenin lipoic acid ester with antioxidant activity and a method of preparing the same | Invented BER lipoic acid ester having excellent antioxidant potential. | [48] |
| 8. | US 8,007,837 B2 | Herbal composition for maintaining/caring the skin around the eye, methods of preparing the same and uses thereof | Invented a novel herbal skinceutical composition to maintain and improve skin health especially for delicate skin around the eyes comprising the extracts of *Saxifraga ligulata*, *Cipadessa baccifera* and *Emblica officinalis,* method for preparing the same and use | [49] |
| 9. | US 7,785,637 B2 | Herbal composition for other publications maintaining/caring the skin around the eye, methods of preparing the same and uses thereof | Invented a novel herbal skinceutical composition to improve and maintain skin health especially for delicate skin around the eyes comprising the extracts of *Saxifraga ligulata*, *Cipadessa baccifera* and *Emblica officinalis,* method for preparing the same and their uses. | [50] |

**Table 3.** *Cont.*

| Sr. No. | Patents No./Patent Publication No. | Title | Invention | References |
|---------|-----------------------------------|-------|-----------|------------|
| 10. | 217147 | A Pharmaceutical Composition Useful as an Antioxidant | Invented the process of isolation of BER from *Tinospora crispa.* | [51] |
| 11. | US 2004/O115286A1 | Cosmetic composition of Remedying skin wrinkles comprising bergenia Emeensis extract as active ingredient | Invented a cosmetic composition having Bergenia emeiensis extract, for skin wrinkles owing to its potential inhibition of collagenase and elastase. | [52] |
| 12. | WO2019077620A1 | Gastroretentive sustained release formulations of *Bergenia ciliata* | Invented novel gastroretentive swellable oral formulations for sustained or delayed release of BER-rich *Bergenia ciliata* extract/fraction and a process for preparing the same. The novel formulations were found to be retained in the stomach, which avoids intestinal degradation of BER resulting in its sustained release in stomach over a time period of 16–24 h. | [53] |

### 3.1. Antioxidant

Antioxidant activity is the efficiency of an active molecule to decrease free radical production and scavenge ROS, repairing, as well as inhibiting, injuries occuring due to the degradation and oxidation of biomolecules and other molecules [54]. BER manifests its antioxidant action by reducing free radical formation and scavenging the various ROS formed. This action of BER has been observed via a strong scavenging effect on DPPH (2,2-diphenyl-2-picryl hydrazyl free radical) in the literature. It also prevents lipids peroxidation [2,22].

Rastogi et al. reported the good antioxidant effect of BER and extract of *Sacoglottis gabonensis* bark. DPPH, ABTS (2,2′-Azinobis-(3-Ethylbenzthiazolin-6-Sulfonic Acid), hydrogen peroxide assays, and lipid peroxidation assays were performed to investigate this effect [43]. The antioxidant potential of aqueous and methanolic extracts of *B. ciliata* was also examined by Rajkumar and his coworkers. Both extracts displayed free radical scavenging activity, which might inhibit oxidative injury [54]. Nazir et al. illustrated that bergenin pentacetate (a peracetate derivative of BER) isolated from *Bergenia stracheyi*, displayed higher DPPH radical scavenging activity [2].

### 3.2. Anti-Inflammatory

Additionally, another well documented property of BER is anti-inflammatory activity [55]. Inflammation is an essential division of native immunity. It includes the removal of damaging signals and the beginning of defensive responses and tissue mending processes in the body. The inflammatory signal is a vital physiological progression that regulates the homeostasis of the immune system. It is segregated into chronic and acute inflammation; both have an important effect on the health of human beings. The acute inflammation progression is evaluated by the rapid employment of granulocytes (eosinophils, basophils, and neutrophils) in the human body. Ailments of sustained inflammatory ablation or stimulation phases can result in chronic inflammation, together with atherosclerosis, silicosis, inflammatory bowel disease, systemic lupus erythematosus, and rheumatoid arthritis [56]. An inflammatory reaction is proliferated by different signaling pathways, mainly NF-κB/P2 × 7/MAPK. Nuclear factor-kappa B (NF-κB) is recognized as a key target for the management of inflammatory disorders. Lipopolysaccharide (LPS) (as a typical activator of the NF-κB signaling pathway) induces a huge quantity of inflammatory factors and triggers inflammatory reactions, furthermore, leading to inflammatory damage in living organisms. Recently, in documented research, BER prevented inflammatory processes by inhibiting the lipopolysaccharide-induced formation of pro-inflammatory cytokines in THP- cells, with selectivity toward IL-6 (interleukin-6) [18]. BER could potently downregulate the formation of Th1 cytokines (IL-4, IFN-γ and TNF-α), and upregulate the formation of Th2 cytokines (IL-5 and IL-4) [5]. A study showed that BER inhibits the release of TNF-α and IL-1β, and significantly suppresses the overexpression of COX-1, COX-2 and PLA2 [9].

BER has anti-inflammatory potential in the management of chronic gastritis and chronic bronchitis [57]. Nevertheless, its pharmacological action on lipopolysaccharide-induced acute lung injury (LPS induced ALI) and its prospective mechanisms are still not clear. To understand anti-inflammatory effects of this molecule, Yang and the research team evaluated its influence on the LPS-induced ALI murine model and explored its prospective mechanism of actions as well. BER declined inflammatory cells, IL-1β and IL-6 formation in bronchoalveolar lavage fluid, and IL-1β, TNF-α, and IL-6 formation in the serum of LPS-induced ALI mice. Moreover, it clearly prevented LPS-induced NF-κB p65 phosphorylation, and MyD88 expression as well. In addition, in RAW 264.7 cells, BER also remarkably impeded the nuclear translocation and NF-κB p65 phosphorylation stimulated by LPS [58].

Bharate and his research group carried out a pre-clinical study of *B. ciliata* extract (IIIM-160; with BER as main constituent; 9.1% *w/w*). IIIM-160 prevented lipopolysaccharide-induced cytokines (proinflammatory) formation in THP-1 (human leukemia monocytic) cells, and selective for IL-6, and showed an outstanding safety index. It displayed antinoci-

ceptive, anti-arthritic and anti-inflammatory actions in several animal models and was found safer up to 2 g/kg (oral dose) in Swiss albino mice [18].

To explore the role of BER in inflammatory mediators [prostaglandin E2 (PGE2) and histamine], Oliveria and his group examined BER (obtained from *PeltophorumdubiumTaub)*, in a carrageenan-induced edema model, and inflammatory mediators in Swiss mice. The outcomes stated that BER possesses anti-inflammatory potential, decreasing the migration of neutrophil and injury occurring due to lipid peroxidation and oxidative stress. No side effects were found due to BER in examined parameters [55].

Chen and his research team investigated the anti-inflammatory mechanism of BER in TNF-$\alpha$-induced human bronchial epithelial (16-HBE) cell lines. The results revealed that BER suppressed TNF-$\alpha$-stimulated proinflammatory signals by enhancing IRT1 action to block the NF-$\kappa$B signaling pathway that may be beneficial for airway inflammation allied with asthma [59].

### 3.3. Anti-Microbial

Infectious disorders respond to very few, or sometimes none, of the available synthetic drugs, and may develop resistance. The progress of novel formulations of natural antimicrobial agents can be of great significance in handling these issues [60]. The antimicrobial potential of BER was assessed against *Salmonella enteriditis, Escherichia coli, Enterococcus faecalis, Pseudomonas aeruginosa, Candida albicans, Staphylococcus aureus, Candida guilliermondii, Aspergillusflavus, A. nidulans* and *C. tropicalis, A. niger, Shigellasonnei, Serratia marcenses* and *Klebsiella pneumoniae*. BER was found to reduce the proliferation of the *C. guilliermondii, C. tropicalis*, and *C. albicans,* however, less activity was observed against *Aspergillus niger, A. flavus*, and *A. nidulans*, and no activity against gram negative and gram positive bacteria [9].

Nazir et al. assessed the antimicrobial and oxidant potential of Bergenin pentacetate [2]. In the year after, Raj et al. assessed the antifungal and antibacterial potential of the methanol extract (crude) of *Peltophorum pterocarpum* flower. It displayed antifungal potential against *Epidermophyton floccosum, Trichophyton mentagrophytes, Botrytis cinerea, Aspergillus niger*, and *Trichophyton rubrum* [61]. In another study, Malik and co-workers prepared a gel composed of the extracts of *Butea monosperma* flowers and *Bergenia ligulata* rhizomes that would help in healingof wounds through antibacterial action at the wound site. Both extracts displayed a maximal zone of inhibition [16].

### 3.4. Anti-Cancer

BER exhibits anticancer potential owing to its antitumor promotion by reducing ROS, DNA fragmentation, $G_0/G_1$ cell cycle arrest, and apoptosis induction. The induction of apoptosis has been linked to augmentation in the expression of Bax and the suppression of the Bcl-2 protein. Furthermore, BER could also inhibit the galectin-3 enzyme, as well as the phosphorylation of $STAT_3$ [19].

Shi and research group evaluated anticancer efficacy of BER on HeLa cervical cancer cells. The findings displayed that BER decreased the cell viability of the HeLa cells in a dose-dependent pattern. On the other hand, the anticancer effects of this bioactive molecule were found to be fewer on the normal cervical cells comparatively. Moreover, the anticancer effects of BER were mainly attributed to the induction of apoptosis in the HeLa cervical cancer cells. In particular, BER also augmented the Bax expression, while it decreased Bcl-2 expression. The effect of BER on cell cycle phase distribution in HeLa cells was also examined, and it was demonstrated that it could provoke $G_0/G_1$ cell cycle arrest. In addition, BER was also found to prevent the migration of HeLa cancer cells and the phosphorylation of $STAT_3$ [17].

Bladder carcinoma is an ordinary malignant tumor in the urinary system with a high morbidity and mortality rate. Liu and his research team examined the influence of BER on bladder cancer proliferation and its associated mechanism. BER was found to reduce bladder cancer proliferation by activating the PPAR$\gamma$/PTEN/Akt signal pathway in this

investigation. Hence, BER may be proposed as a promising therapeutic biomolecule for the management of bladder carcinoma [62].

Jayakody and his research team examined the possible mechanism of action of BER. This team postulated anticancer property via computational tools as well. The molecular docking of findings presented that BER: galectin-3 complex is extremely stable, and evidenced the veracity of the docking outcomes, which recommended the potential inhibitory effect of BER on galectin-3. This research confirmed that BER could significantly be employed for the development of potent galectin-3 inhibitors. This study also provided scientific validation for the employment of BER rich plants in the management of cancer in Eastern traditional medicine [63].

### 3.5. Anti-Diabetic

Diabetes mellitus, a persistent metabolic disorder, takes place when the human body is unable to generate sufficient insulin, or utilize insulin efficiently, or both. Employing drugs (DPP-4 inhibitors, GLP-1 agonists, SGLT-2 inhibitors and metformin) controls glucose levels between 70 and 99 mg/dL, however, unfavorable side effects are observed when it comes to the usage of these drugs. Nowadays, researchers have shown interest in BER, owing to its anti-diabetic efficacy, without any side effects [64]. BER inhibits the α-glucosidase enzyme and has a positive effect on the endocrine cells of the pancreas, leading to augmented insulin formation [22].

Kumar and his research group isolated BER from the roots of *Caesalpiniadigyna* Rottler (Leguminosae) and examined it for anti-diabetic potential in type 2 diabetes in streptozotocin (STZ) nicotinamide-induced diabetic rats. The outcomes of the study recommended that isolated BER has promising antioxidant, hypolipidemic, and anti-diabetic activity in diabetic animals [23].

Recently, Rajput et al. evaluated the effects of BER in beta cells in the presence of cytokines. This bioactive potentially prevented beta cell apoptosis, due to a decline in caspase-3 activity, and concurrently enhanced cellular ATP Levels. BER was also found to remarkably augment insulin secretion in INS-1E cells, most probably due to reduced nitric oxide formation. Furthermore, BER restored mitochondrial membrane potential, decreased ROS formation, and enhanced mitochondrial dehydrogenase potential. This study showed that BER sheltered beta cells from cytokine-induced apoptosis and restored insulin secretory functions, owing to its anti-apoptotic, antioxidant, and anti-inflammatory features [65]. In a nutshell, the abovementioned information highlights the promising potential of BER in the context of diabetes.

### 3.6. Neuroprotective

With an increase in day-to-day social life pressure and aging, people are more prone to the risk of brain ailments, such as depression, Parkinson's disease (PD), epilepsy, and Alzheimer's disease (AD). BER plays a major part in the management of brain and neuron diseases, predominantly related to the central nervous system [66]. Earlier studies have revealed that BER possesses in vitro inhibitory activities against mushroom tyrosinase, bovine adrenal tyrosine hydroxylase, the β-secretase (BACE-1) enzyme, and neuronal death in a primary culture of rat cortical neurons. This herbal compound has also been proposed to prevent human protein tyrosine phosphatase-1B (hPTP1B) in vitro, which is a major target in neuro-inflammation for anti-cancer and anti-diabetic molecules. The investigation carried out by Baria et al. has also pinpointed the ameliorative and preventive potential of BER for Alziemer's disease via multiple targets [67].

Parkinson's disease (PD) is the most commonly occurring neurodegenerative disorder, with cognitive impairment and motor ailments. Ji et al. examined BER (isolated from *Saxifrage stolonifera* Curt. herb (Hu-Er-Cao)) effects on PD, and involved essential mechanisms. In vitro, BER reduced the cytotoxic effects on PC12 cells due to the culture supernatants (obtained from LPS-induced microglial cells). Furthermore, BER reduced PD manifestation through the activation of PI3K/Akt signaling pathway in vitro, as well as in vivo [66].

BER was also screened via a molecular docking study to predict its potential against targets of AD (butyrylcholinesterase (BuChE) (1P0I), acetylcholinesterase (AChE) (1B41), BACE-1 (1FKN) and Tau protein kinase 1 (GSK-3β) (1J1B). The fitness and GOLD score of BER were found to be comparable to positive controls (donepezil, physostigmine, galanthamine). In a nutshell, the experiments have been found to describe the cholinesterase inhibitory potential of BER because of its Aβ-1-42 reduction and p-tau levels, anti-inflammatory activity, and antioxidant effects [67]. The outcomes of the investigation indicated the potential of BER in brain disorder management.

### 3.7. Hepatoprotective

Liver is actively involved in the metabolism and detoxification of different endogenous and exogenous molecules. Several chemicals and drugs, such as pesticides, alcohol, industrial reagents, and carbon tetrachloride, tend to cause liver disorders [68]. BER exerted hepatoprotective potential by attenuating the actions of alanine aminotransferase, sorbitol dehydrogenase, aspartate aminotransferase, γ-glutamyl transferase, inhibiting lipid peroxidation, and recovering the reduced hepatic glutathione level [25]. The hepatoprotective potential of BER (main component of *Mallotusjaponicus)*, was examined in carbon tetrachloride-induced liver injury in rats. This investigation evidently indicated that BER has a significant liver protective potential against carbon tetrachloride-induced liver injury in rats [25].

Xiang and co-workers explored the defensive features of BER on hepatic ischemia reperfusion (IR), predominantly the purging of ROS and activation of PPAR-γ (peroxisome proliferators activated receptor γ). This compound exhibited liver protection via ROS-influenced inflammatory factors release, and autophagy- and apoptosis-associated genes through the PPAR-γ pathway in this hepatic IR injury model [69].

Liver fibrosis is another pathology which involves the diffusion of extracellular matrix (ECM) deposits. In an investigation, Xia and his research team assessed the defensive effect of BER on hepatic fibrosis via bile duct ligation and carbon tetrachloride. In this study, BER was observed as a potential bioactive for the management of hepatic fibrosis [70].

Sriset et al. inspected the hepatoprotective potential of BER (a natural derivative of gallic acid) against tert-butyl hydroperoxide (TBHP), and ethanol-induced oxidative damage in human hepatoma (HepG2) cell lines. The results of this study demonstrated that BER possessed a liver-protective property through the re-establishment of the oxidant–antioxidant system, and hence, is a promising molecule for liver protection [71].

### 3.8. Anti-Hyperuricemic

Uric acid is a metabolic product (end product of purine metabolism) that has no physiological role in our body. Hyperuricemia is a condition in which an elevated level of uric acid is commonly observed in blood. This disease is a result of augmentation in metabolism, a decrease in gout or under excretion, or a combination of both. It can lead to gout, an inflammatory disorder, which occurs due to the formation of uronatrium crystals in certain tissues and joints [72].

Recently, Chen and his research team investigated the potential utility of BER molecules in hyperuricemia. BER was found to diminish serum urate levels in hyperuricemia-induced mice by facilitating gut and renal uric acid excretion. This therapy augmented the expression of Abcg2, both in intestines and kidneys, whereas Slc2a9 expression was found concealed in kidneys and augmented in intestines. Furthermore, BER promoted ABCG2 expression in HK-2 and Caco-2 cells, and SLC2A9 in Caco-2 cells as well, by activating PPARγ. However, BER weakened the expression of SLC2A9 in HK-2 cells by preventing the nuclear translocation of p53. Moreover, BER was also found to reduce TNF-α, IL-6, and IL-1β levels in serum in hyperuricemia-induced mice. In a nutshell, the findings supported the promising pharmacological effect of BER in the management of hyperuricemia [59].

### 3.9. Immunomodulatory

Qi and his team investigated the effect of BER on the immune system and antioxidant effects in immunosuppressed rodents (mice). In this study, the estimation influence of BER on the immune system was assessed. A histological examination and indexes of immune organs found that cyclophosphamide (Cy) exhibited thymus and spleen injury in comparison to normal control, which was reduced by BER. Furthermore, BER also augmented the humoral immune function by enhancing serum IgM and IgG levels. An enhancement in the cellular immune function was also observed. The findings indicated that BER enhanced the proliferation of B and T lymphocytes, peritoneal macrophage functions, T (CD8+ and CD4+) lymphocyte subsets, CTL, and NK cell activities. Moreover, BER inverted the Cy-induced decline in the total antioxidant efficiency involving catalase (CAT), glutathione peroxidase (GSH-Px), and superoxide dismutase (SOD) processes. In a nutshell, BER sheltered mice against Cy-induced unfavorable reactions by augmenting cellular and humoral immune functions, and enhancing antioxidative potential [73].

## 4. Applications

Various applications of bergenin are elaborated previously in Table 1.

### 4.1. Anxiolytic

Anxiety ailment is the most common mental ailment faced by adolescents and children. Prevalence rates of anxiety are from 13.6 to 28.8% in western nations and 4.5% of the global population. The anti-anxiety potential of BER was examined by Singh and his co-workers. BER has been shown to have remarkable anti-anxiety potential (at 80 mg/kg, per oral), i.e., statistically similar to diazepam (2 mg/kg/per oral). BER exhibited significant anxiolytic potential in mirrored chamber and open field tests as well [74].

### 4.2. Antimalarial

Malaria is a severe protozoal parasitic disease transmitted by Anopheles mosquitoes (female) [27]. As per the literature evidence, BER is highly active both for chloroquine-sensitive (CQS) and chloroquine-resistant (CQR) P. *falciparum*. BER represents its action through generating oxidative stress and via preventing hemozoin formation, and therefore provokes the decease of malaria parasites [75].

Uddin et al. have assessed anti-plasmodial properties of BER in comparison to 11-O-galloylbergenin (its natural derivative). Both constituents were collected from *Bergenia ligulata* [21]. Liang and researchers also examined the antimalarial activity of BER (derived from *Rodgersiaaes culifolia* Batal). BER successfully prevented the in vitro growth of *P. falciparum*, besides apparent cytotoxic to mammalian HepG2 and HeLa cell line or to erythrocytes. BER administration to *Plasmodium berghei*-infected mice for 6 days remarkably prevented the extension of the parasites [27]. Singh and co-workers bioprospected leaves of *Flueggeavirosa* for its anti-malarial efficiency and active principles. BER showed modest anti-malarial action against *P. berghei* and reduced parasites causing systemic inflammation in mice as well [76]. In a nutshell, these outcomes substantiated that BER is a potential bioactive molecule for the management of malaria.

### 4.3. Antituberculosis

Tuberculosis is one of the most global health concerns, which has delayed socio-economic progress in various areas of the world [28].

Dwivedi and their research group showed that BER (from tender leaves of *Shorearobusta*) activates ERK and MAP kinase pathways and stimulates NO, IL-12 and TNF-$\alpha$ formation in infected types of macrophages. Furthermore, BER stimulates Th1 immune signals and potentially prevents bacillary growth in a mycobacterium tuberculosis-infected murine model. These findings identified BER as a potential bioactive for TB management [28].

Kumar and his research team found that this bioactive candidate stimulates T helper 17 (Th17)- and Th1 cellular defensive immune responses, and potentially prevents the growth of mycobacterials in the mycobacterium tuberculosis-infected murine model. Of note, BER treatment remarkably declined the bacterial burden of an MDR TB strain. These outcomes demonstrated that BER is a powerful immunomodulatory drug candidate which can be explored as a prospective adjunct to TB treatment in near future [77].

### 4.4. Antiplatelet Aggregation

Thrombosis is the biggest cause of death in the world. This disorder is intimately associated to a chain of cascades such as secretory, aggregation and adhesive purposes of the activated platelet, and the activation of the extrinsic and intrinsic coagulation systems, which are accountable for fibrin formation and blood coagulation. In particular, the aggregation of platelets significantly participated in several thromboembolic ailments as well [78].

Alkadi and his research team isolated four known constituents, 5-Hydroxyflavone, 2′-Hydroxyflavanone, Paeonol, and BER, from the stem bark of *Garciniamalaccensis*. All isolated components were found to exhibit the prevention of platelet aggregation in human blood, stimulated by collagen, ADP (adenosine diphosphate), and AA (arachidonic acid) [3].

### 4.5. Antihyperlipidemic

Hyperlipidemia contributes potentially to the manifestation and development of atherosclerosis and coronary heart diseases (CHDs) [79].

Jahromi and his co-workers stated that BER (collected from the leaves of *Flueggeamicrocarpa)*, administered through the oral route to hyper-lipidaemic rats, remarkably, lowered total lipid in serum devoid of much variation in serum triglycerides and cholesterol. After 21 days of oral administration, triglycerides, cholesterol, very low-density lipoprotein (VLDL)-cholesterol, and low-density lipoprotein (LDL) levels in serum were potentially decreased, whereas high density lipoprotein (HDL)-cholesterol levels in serum were found to be enhanced. Animals treated with BER presented a remarkable reduction in atherogenic index as well [29].

### 4.6. Anti-Arrhythmic

Dysrhythmias or cardiac arrhythmias are disagreements in the electrical action of the heart, including unbalanced rhythm or a rate known as bradycardia or tachycardia, respectively. Commonly, the mechanism of action underlying cardiac arrhythmias may be divided into two classes: abnormal impulse propagation, abnormal impulse generation, or both [80].

Pu et al. isolated BER from the aerial parts of *Fluggeavirosa* (Euphorbiaceae), and investigated its anti-arrhythmic potential. The observations suggested that BER possesses a potential property to treat cardiac arrhythmias [31].

### 4.7. Anti-Ulcer

An idiopathic inflammatory bowel ailment, ulcerative colitis is characterized by the inflammation of intestine; the inhibition of this inflammatory process may become a method for the evolution of novel anti-inflammatory bioactives, with higher efficacy and minimum adverse effects [55]. Ulcer is a major ailment of the gastrointestinal system which influences 10% of the world population, with different etiologies [81].

Goel et al. demonstrated norbergenin, BER (extracted from the roots and leaves of *Flueggeamicrocarpa*), and luvangetin, (extracted from the seeds of *Aeglemarmelos Correa*). These presented a remarkable defense in aspirin-induced gastric ulcers in rats and cold restraint stress-induced gastric ulcers in guinea pigs and rats on oral administration. The findings revealed that the gastro-defensive behavior of BER and norbergenin was due to enhanced prostaglandin formation, whereas some of different mucosal protective parameters can be engaged with luvangetin as well [32].

Oliveira and his research team explored the role of BER in the acute colitis induced by 2,4,6-trinitrobenzenesulfonic acid (TNBS) in rat models. The results demonstrated that BER reduces the microscopic and macroscopic injury symbols of colitis, and neutrophilic infiltration level in the colonic tissue. Furthermore, it was able to downregulate pSTAT3 protein expression, IkB-$\alpha$, iNOS, and COX-2. This study has substantiated that BER decreases the injury caused by TNBS in acute colitis rat models and cytokines and proinflammatory proteins levels, possibly via the modulation of NF-$\kappa$B and pSTAT3 signaling, and inhibiting non-canonical and canonical NLRP3/ASC inflammation pathways [55].

In an investigation, Wang and his research group isolated BER from the herb of *Saxifragastolonifera Curt.* (Hu-Er-Cao), and identified its effect on investigational colitis, and associated mechanisms. The outcomes displayed that its oral administration potentially reduces disease manifestations in mice, ascertained by MPO activity, the shortening of colon length, and decreased disease activity index scores and pathological abnormalities in colons [82].

### 4.8. Wound Healing

For a long time, several parts of the Indian ethnomedicinal plant "*Shorearobusta*" have been conventionally employed for various disorders, such as burns and wounds, by various tribal communities. The animals treated with the fractions (5%) and extracts presented potential minimization in the wound area (96.41% and 96.55%), with faster epithelialisation (17.86 and 17.50), whereas the isolated constituents ursolic acid and BER heal the injury more rapidly. Furthermore, the hydroxyproline content, granuloma tissue weight, and tensile strength of the incision wound were potentially enhanced by both the compound(s). Moreover, as per tissue histology outcomes, isolated compound(s) presented entire epithelialization with enhanced collagenation, comparable to povidone-iodine [83].

### 4.9. Bone Healing

Suh and research group studied the BER effect on osteoblasts (MC3T3-E1). BER therapy potentially raised the synthesis of osteocalcin, action of alkaline phosphatase, synthesis of collagen fibers, and cells mineralization. Moreover, BER therapy remarkably suppressed the transcription factor 6 activation and MG (methylglyoxal) autophagy. The outcomes represented that BER may have good efficacy when it comes to functions of osteoblastic cell [84]. MG is the main precursor for the synthesis of AGEs (advanced glycation end products). Pretreating cells (MC3T3-E1) with BER inhibited the formation of protein adduct induced by MG. BER suppressed the soluble receptor for the interleukin, AGE (sRAGE), superoxide, and ROS making induced by MG. In addition, in the presence of MG, BER enhanced heme oxygenase-1 glutathione, nuclear factor erythroid 2-related factor 2 levels, and glyoxalase I activity. The findings showed that BER may be a good bioactive for the management of diabetic osteopathy [85].

Bone mesenchymal stem cells (BMSCs) are vital applicants for the regeneration of bone. Hou and his research group examined the potential of BER on the BMSCs osteogenesis, and an in vitro study showed that BER augmented osteoblast-specific markers and suppressed the adipocyte-specific markers. These outcomes presented that BER enhanced the differentiation of osteogenic of BMSCs, at least partly via SIRT1 activation [86].

### 4.10. Anti-Bronchitis

Nowadays, air pollution is increasing dramatically, which leads to diseases like CB (chronic bronchitis), becoming a cause for grave concern worldwide. As per the literature evidence, numerous studies have presented the significant utility of BER for chronic bronchitis therapy. To explore this bioactive for the treatment of CB, Zhang and the research group investigated BER mechanism in CB by employing a serum metabolomics method. This bioactive was given orally to rats after exposure. It was found that BER inhibited inflammatory cells infiltration, mucus secretion, and white blood cells. The findings of this study showed that the therapeutic action of this bioactive compound may be asso-

ciated with the management of glycerophospholipid, arginine, tryptophan, and proline metabolism altered in CB [87].

Chen and co-workers examined the anti-inflammatory potential and mechanism of BER in TNF-$\alpha$-stimulated human bronchial epithelial (16-HBE) cells. The results suggested that BER could circumvent TNF-$\alpha$-induced proinflammatory results via augmenting the SIRT1 [sirtuin (silent mating type information regulation 2 homolog) 1] mechanism to inhibit the NF-$\kappa$B signaling pathway, which may provide advantageous potential to cure the inflammation of the airway related to asthma [59].

### 4.11. Antileishmania

Medicinal bioactives with immunomodulatory activities can offer better alternative therapeutic approaches for the treatment of visceral leishmaniasis. BER is an interesting phytoconstituent with strong antimicrobial, antioxidant, and immunomodulatory potential. Kaur and her research team determined the antileishmanial potential of BER by modifying the immune system responses of BALB/c mice. The therapeutic action was examined via in vitro antileishmanial assay, and in vivo by using BALB/c mice. This research team demonstrated the good potential of BER against immunomodulatory and antileishmanial disorders [88].

### 5. Bergenin Novel Formulations

BER has shown notable activity against an extensive array of chronic disorders. Nevertheless, the pharmaceutical importance and therapeutic efficiency of BER is limited, owing to its poor oral bioavailability [89], lower aqueous solubility, and permeability [19]. To surmount these pharmaceutical hindrances, new formulation strategies need to be developed. In recent times, the scientific community has been focused on the fabrication of nano-based carrier systems. These novel strategies include phospholipid complex, extended-release core tablets, prodrug, herbal gel, phospholipid complex solid dispersion, poly herbal ointment, nanoparticles, and poly lactic acid polymers [90]. Considering the limitations, researchers have designed various drug delivery systems to enhance its therapeutic efficacy; each of these systems is deliberated in detail (Table 4) in the below sections.

### 5.1. Phospholipid Complex

To enhance the oral bioavailability of BER, phospholipid complexes (PC) were prepared of this bioactive. For the optimization of the bergenin phospholipid complexes (BPC) fabrication technique, a spherical symmetric design response surface method was employed. The formulation was well fabricated in the optimal conditions, such as 60 °C temperature, 80 g/L concentration of the drug, and the 0.9 ($w/w$) ratio of the drug to phospholipids. The combination percentage was found to be $100.00 \pm 0.20\%$, and BER content in the formulation was found to be $45.98 \pm 1.12\%$. The BPC solubility in n-octanol and water was efficiently increased. The value of $C_{max}$ and $AUC_{0\to\infty}$ of prepared formulation was found to be improved, and the relative bioavailability of the drug was successfully enhanced (439% of BER). Hence, it was concluded that BER PC is an appreciated carrier system to increase its absorption orally [89].

In another research, Guan et al. examined the mechanism of augmentation on BER oral absorption by employing the BER–phospholipid complex. Various models involving the everted rat gut sac (ex vivo) and Caco-2 cell (in vitro) were employed. The outcomes from the ex vivo model of everted rat gut sac concluded that the small intestine is more convenient for this bioactive absorption with respect to the colon. Furthermore, the chitosan addition could open the intestinal epithelial cells tight junctions, thus efficiently enhancing the BER transport through the paracellular route. Hence, the results of this study would be worthy of the future evolution of oral carrier systems of BER [91].

### 5.2. Phospholipid Complex Solid Dispersion

BER is a BCS IV class bioactive, and potentially removed by the P-glycoprotein (P-gp) efflux function. Thus, BER–phospholipid complex solid dispersion (BERPC-SD) was fabricated by employing the solvent evaporation method and evaluated. BER and BERPC-SD alone, and along with verapamil (P-gp inhibitor), were given to Sprague–Dawley rats to check the P-gp effect on BER absorption in the intestine. It was also found that the membrane permeability of BER bioactive moiety from BERPC-SD was more than BER alone, and was enhanced further by co-administration with verapamil. A pharmacokinetics assay was conducted in Sprague–Dawley rats to find out the BER level in plasma. The BER level in plasma was estimated via high performance liquid chromatography. The values of $C_{max}$ and $AUC^{0-t}$ for this bioactive were significantly greater in BERPC-SD then pure bioactive, and were further augmented by verapamil. These findings represent that BERPC-SD can upsurge the bioavailability of BCS class IV moieties [92].

### 5.3. Coated Floating Tablets

A novel coated gastric floating drug delivery system (GFDDS) of BER and cetirizine dihydrochloride (CET) was fabricated. Firstly, the pharmacodynamic assays were carried out, and the findings disclosed that the novel formulation of BER/CET was found to be comparatively more efficacious with respect to commercial products, i.e., BER/chlorphenamine maleate, but with lesser adverse effects on CNS. Consequently, extended release core tablets of BER were prepared, whereas CET was amalgamated into the gastric coating film for instant release. This novel formulation offers a new delivery approach to improve BER absorption at its specific site, and the efficiency of both drugs [93].

### 5.4. Prodrug

BER is an exceptional C-glycoside herbal compound possessing good anti-inflammatory as well as anti-arthritic potential. It is a lyophobic moiety and is not stable at neutral basic pH. The degradation rate is directly linked to pH enhancement, which may be one of the main reasons for its less oral absorption. Thus, Singh et al. improved the stability of this bioactive employing prodrug strategy. The stability of prepared prodrugs was evaluated in buffers at various pH, as well as in the bio-relevant medium. All prodrugs showed significant enhancement in the lipophilicity of the drug. It was concluded that acetyl ester may be the best promising prodrug, as it was more stable at gastric or intestinal pH and was entirely changed to the BER in plasma. The findings of this research may help in fabricating the stable prodrugs of unstable moieties with anticipated physicochemical characteristics [7].

### 5.5. Herbal Gel

Malik and his research group fabricated a gel containing both extracts of the rhizomes of *Bergenia ligulata* and flowers of *Butea monosperma* that might aid in wound healing due to their antibacterial effect on wounds. Soframycin cream and the gel were used as standard for comparison. The antibacterial assay was conducted using the agar well-diffusion study, at odds with *S. aureus, Pr. Vulgaris*, *MRSA*, and *E. coli*. With this assay, both extracts (concentration about 100 μg/mL) demonstrated their highest inhibitory zones. This concentration of their experiment was useful in calculating the topical gel formulations dose of BER [16].

**Table 4.** Formulations reported in literature for Bergenin.

| Sr. No. | Formulations | BER Concentration | Techniques | Applications | Assay | References |
|---|---|---|---|---|---|---|
| 1. | Phospholipid complex | - | Solvent evaporation method | Increased oral absorption | In vitro and in vivo | [89] |
| 2. | Coated floating tablets | 187.5 mg | Single-punch machine | Prolonged gastric retention time | In vitro and in vivo | [93] |
| 3. | Prodrug | - | - | Increased stability | | [94] |
| 4. | Herbal gel (Aqueous Extract of *Bergenia Ligulata* Rhizomes and Ethanolic Extract of *Butea Monosperma* Flowers) | - | Soxhlet extraction, blending | Wound healing by exhibiting antibacterial activity at the site of wound infection | In vitro | [16] |
| 5. | Phospholipid complex solid dispersion | - | Solvent evaporation | Increase the oral BCS IV drugs bioavailability | In vitro and in vivo | [92] |
| 6. | Poly herbal ointment (*Bergenia Ciliata*) | - | - | Excision and incision wounds | - | [6] |
| 7. | Nanoparticles | 0.5 mg/mL | - | Enhancing physiochemical properties and anti-arthritic activity | In vivo | [19] |
| 8. | Sustained release capsules (*Bergenia Ciliata*) | - | - | Sustained release | - | [18] |
| 9. | Poly lactic acid polymers | | - | Sustained release | In vitro | [95] |

### 5.6. Nanoparticles

Rao and the research team fabricated BER-loaded silver nanoparticles (AgNPs) stabilized with gum xanthan for anti-arthritic action in a complete Freund's adjuvant-induced arthritis model. Oral administration of BER loaded nanoformulation exhibited good anti-arthritic potential with a lower score of arthritis, mild to moderate swelling of the paw tissue, decreased degenerative differences including the moderate influx of inflammatory cells and mild articular changes. The administration of this bioactive and its NPs significantly inhibited the ROS levels when compared to the control group. Furthermore, enhanced $O_2^-$ production in neutrophils was also inhibited. The antioxidant potential was further assayed by their inhibitory action against TLRs (Toll-like receptors) and cytokine production. This study concluded that optimized gum xanthan-stabilized AgNPs are stable, and possess the efficient multi-targeted delivery of BER with the promising management of arthritis [19].

### 5.7. Poly (Lactic Acid) Polymer

From the literature, it was found that the efficacy of BER fabricated with osmotic-pump-controlled release is lesser than expected. In a research, PLA (poly (lactic acid) (biodegradable) was employed to alter BER and stabilize this bioactive with the chemicals technique. BER-PLA produced by this technique has a lower molecular weight and better thermal stability, along with extended in vitro release, as well as enhanced molecular weight. An in vitro antitumor and biocompatibility assay exhibited that BER-PLA (1:30 ratio) has enhanced biological activities and lower cytotoxicity, and its anti-cancer action was significantly enhanced compared to the pure bioactive molecule [96]. Furthermore, in 2020, Ren and his team also modified BER to combine with amino poly (lactic acid) for the chemical immobilization of this drug. The fabricated BER-amino PLA was controllable with a lower molecular weight and superb stability. It was found that in the in vitro biological compatibility and anti-tumor assay, BER-amino PLA was found to have promising biological compatibility and lower cytotoxicity compared to the standard group. It was demonstrated that the chemical immobilization of BER not only provides a great method of administration of the drug, but paves a noble way for the prolonged release of this compound [95].

### 5.8. Sustained Release Capsule

A research group performed a preclinical evaluation of the IIIM-160 (*Bergenia ciliate*-based botanical extract has a 9.1% *w/w* of BER). The in vitro and ex vivo potential were evaluated by using inflammation models, nociception and arthritis. An acute oral toxicity assay was conducted on Swiss albino mice. A suitable oral formulation (sustained release capsules) was prepared and evaluated. It showed the inhibition of pro-inflammatory cytokines in THP-1 cells, particularly towards interleukin-6, and with a good safety window. It exhibited anti-inflammatory, antinociceptive, and anti-arthritic action in animal models, and was found to be non-toxic in Swiss mice at up to 2 g/kg oral doses. The gastroretentive capsules displayed an extended release of BER over a time period of 24 h [18].

## 6. Pharmacokinetics

Pharmacokinetic evaluations not only play a significant role in disclosing the effectiveness of natural drugs, but can also the active constituents. From the pharmacokinetic analysis in the last decade, the intracorporal BER mechanisms were confirmed. The in vivo providence of BER will be determined by the process in the human system under physiological or pathological situations. The bioavailability of BER was calculated by administration of this bioactive at 50 mg/kg and 5 mg/kg, by oral and intravenously route, respectively [18]. After BER 12 mg/kg oral administration, it is entirely excreted in 24 h in bile, and its main excreted quantity was 97.67% in initial 12 h. The 8.97% drug recovery occurs within 24 h in bile. In urine, 95.69% of its quantity was excreted in the initial 24 h, and <22.34% of drug recovery was found within 24 h. 52.51% of total recovery of BER, and

its glucuronide metabolite was observed (32.20% in urine in 48 h and 20.31% in bile in 24 h) [8].

The efficacy of pharmacokinetic assays was described by the calculation of BER concentration in plasma after the administration by IV to rats in doses (30.0, 15.0 and 7.5 mg/kg, taking water as the solvent). The half-lives for elimination and distribution are not associated with the doses administered. A biphasic event with fast distribution along with a sustained elimination segment was assessed through plasma concentration-time graph and follows first order kinetics [97]. The central volume of distribution (Vc) was 0.67 L/kg, while the peripheral volume of distribution (Vp) was 11.35 L/kg. This indicated a wide distribution for BER in rat. The biological half-life (T1/2β) was 4.13 h, which showed that BER exhibited a moderate elimination velocity in rat. In addition, researchers also conducted a pharmacokinetic study of BER administered to rats orally at a 22.5 mg/kg as a single dose. The results indicated that BER was degraded effortlessly in the digestive system, which included a first-pass metabolism, or demonstrated a poor absorption from GIT in rat [98]. After oral administration in humans, bergenin was absorbed quickly but incompletely, with a short half-life and low bioavailability. The $C_{max}$ appeared at 1–4 h in plasma and at 2–7 h in urine after intramuscular administration in dogs (Jiangsu New College Med. 1999). Shi et al. concluded that the low bioavailability might be attributed to the easy degradation of BER in the digestive system. However, no further studies of BER stability after oral administration have been reported [37].

## 7. Conclusions: The Road Ahead

The usage of herbal drugs is continuously escalating, with about 4/5 of the worldwide population still depending on traditional folk remedies for their primary health-related solutions. Herbal products have minor toxicities, which are the main reason for their holistic therapeutic usage. One such holistic bioactive compound is Bergenin, which is endowed with some promising proven therapeutic activities, such as antioxidant, anti-inflammatory, anticancer, anti-infective, anti-diabetic, neuroprotective, anti-urolithiatic, hepatoprotective and anti-hyperuricemic properties. Therefore, this compound has attracted the great interest of scientists in the 21st century. Based on documented evidence, we systematically reviewed the beneficial properties, mechanism of action, chemistry, pharmacokinetics, and drug carrier systems reported for BER, despite general discussion. It can be concluded from the literature that this compound has potential for treating various ailments. Certain explanatory mechanistic studies should also be conducted so as to unravel the underlying mechanisms of action of this compound, thereby validating the traditional knowledge associated with it. However, more clinical research is necessary to investigate the efficacy of BER. Nonetheless, BER is neither a highly hydrophilic nor highly lipophilic compound with poor solubility, permeability, and oral bioavailability, with short half-life and degrading at intestinal pH, which limits its applications. It is worth mentioning here that, besides the widespread utilization of this compound, in traditional medicine, scientific data with regard to its delivery systems are scanty. Hence, it is understandable that newer drug delivery systems should be explored to resolve the abovementioned hitches of BER, without compromising on patient acceptability. These delivery systems enhance the chances of clinical trials of this compound in the near future. The discussion of BER will encourage scientists to design and formulate an ideal drug carrier system. Furthermore, research in this field may lay a foundation for future marketed goods of BER that are therapeutically safe, eco-friendly, and effective. We also anticipate an improvement in patented dosage formulations for BER in the future.

**Author Contributions:** Conceptualization, writing and original draft Preparation: R.R., S.M., V.K. and S.D.; Review and editing: R.R., V.K., S.K., P.D. and M.G. All authors have read and agreed to the published version of the manuscript.

**Funding:** This research received no external funding.

**Institutional Review Board Statement:** Not applicable.

**Informed Consent Statement:** Not applicable.

**Acknowledgments:** The authors, Varsha Kadian, Pooja Dalal sincerely expresses gratitude to Indian Council of Medical Research, New Delhi for providing Senior Research Fellowship (Letter No: 45/55/2020-Nan/BMS) and (Letter No: 45/41/2020-Nan/BMS), respectively.

**Conflicts of Interest:** The authors report no conflict of interest in this work.

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
