# Peer review of "A Fresh Look on Bergenin: Vision of Its Novel Drug Delivery Systems and Pharmacological Activities"

_futurepharmacol, doi:10.3390/futurepharmacol2010006_

Round 1
Reviewer 1 Report
The manuscript is a comprehensive review on Bergenin, a natural compound with several beneficial effects. The authors focused on the design and the development of novel delivery systems for Bergenin and provided literature evidence on both chemistry and pharmacology of the compound. The subject matter is of interest; however, the manuscript is redundant and should be reorganized for the sake of clarity and readability. For example, in section 3 the authors reported the Bergenin mechanism of action, the same evidence was also in the other sections. In addition, a critical appraisal of the cited literature should be performed. I also suggest to carefully revised the English language. Figures should also be renumbered according to their citations in the main text.
Author Response
Sincerest thanks for your response and comments on our manuscript.
We have revised the manuscript carefully to fully address the changes as per suggestions as well as outlined using track changes in response to your comments.
Reviewer 2 Report
This is an interesting review, the study is of maximum topicality, relevant, the authors contribute to this field of research. The results of previous studies are well classified, summarized and discussed.
Must be specified in Introduction, which was the method of selection of the processed articles in the review.
The authors need to check the list of references and correct it, according to the Instructions for authors
Author Response
Sincerest thanks for your response and comments on our manuscript.
We have revised the manuscript carefully to fully address the changes as per suggestions as well as outlined using track changes in response to comments.

Reviewer 3 Report
General comments:
-English requires considerable attention
-there is no logic in the numbering and positioning of the Tables and Figures
- a lot of words are too plastic/less scientific for a scientific review and they need to be replaced. ‘
-some paragraphs have the impression that the review is history book or yearbook.
L9: Delete ‘family’
L10: replace ‘army’ with a less plastic word.
L15: Keep only ‘newer’ or ‘recently’.
L15-17: Needs rephrasing
L34, 38: replace ‘secondary’ with ‘specialized’
L29-31: Sentence needs rephrasing.
L38: the authors should specify which phytochemical class BER belongs to.
L40: ‘Bergenial’ does not exit. Correct
L41& all over the text (including Table 1): It is ‘ciliata’ not ‘ciliate’
L45,47 & all over the text: Make Bergenia in italics.
L54-56: Parts of it are a repetition of Lines40-44. The authors should some rearrange and merge these two paragraphs.
L58: BER is a compound, not a ‘moiety’
L59: Replace ‘miracle’ by a less plastic word.
L60-63: The authors should reduce the length. Just select the most important bioactivities. The sentence is too long and the idea can be lost while reading.
L68: Replace ‘significance’ with ‘use’.
L72, L101, L129, L186, L243 & all over the text.: Replace ‘bergenin’ with ‘BER’
L80-82: I do not see the relevance of this sentence to the current review.
L92: Page 3: I do not understand the numbering of the Figures. Fig. 3 then 6?? Why not Fig. 1 and Fig.2?? In general, the order of the figures and Tables does not make sense
L93: In Fig. 3’; For Bergenia ciliate should be Bergenia ciliata.
L98: Replace “Bergenia species’ by ‘Bergenia spp.’ (and all over Table 1); the family names of the species are not required in this table
L98: Title of the table: These are not applications of Bergenin, but of plants containing bergenin. So rename the title of the Table 1.
L98: In column ‘parts used’ the authors should decide if they use the plural or the singular.
L99: The name of the Table 3 should be ‘Indian commercial formulations for bergenin’
L129-130: Rephrase the sentence. Does not make sense
L135: Transcutol is a brand name. The authors should replace it with the chemical name.
L138-143: They go better together with L116-117.
L145: Section 3 and 4 could be merged. The authors could discuss the mechanisms of action, after describing the individual activities (Subsections) from Section 4.
L193: The authors did not explain the patents in the Text. They just mentioned that they exist and then they provided Table 4.
L201: Table 4 – There is no information from this table that BER or BER-contanining products have been used in this patents. Improve the table
L191: Replace ‘amazing’ with a less plastic word.
L223: From where the abbreviation ‘CY’ comes?
L221-L229: These effects sound to me like more immunomodulatory than pure antioxidant. Maybe the authors should reconsider where to put these lines.
L239-242: The authors do not need to use the brackets shorter names after the name of the species.
L247: Delete ‘in the year 2011’.
L256: Delete ’In the year 2019’
L273: Delete ‘in the year’
L292: Delete ‘clinically’. Alternatively, add the corresponding references for the clinical trials and uses of BER in gastritis and bronchitis and detail those studies, as they would be very important.
L295: Replace ‘mysterious’ with a less plastic word.
L309: Delete ‘in the same year’
L317: Delete ‘in the following year’
L366: Delete ‘in the same year
L378: Delete ín the year 2000
L382: delete ‘in the year 2020’ Keep doing that for all over the text.
L418: Table 5 and Table 1 should be merged into a single table. .
L416: Section 5 should be also the same as Section 3 and 4. I propose to the author to merge all these 3 subsections under Section 3: Pharmacological activities. The mechanisms of actions should be included at their specific action (e.g. antioxidant, anti-inflammatory), after detailing the specific-activitiy studies. Then continue the numbering in logical manner; for instance, after 3.9 Anti-hypeuricemic come 3.10. Anxiolytic, and so on.
L446: Replace with ‘antituberculosis’
L549: Replace with ‘anti-bronchitis’
L587: Table 2 should be improved. The BER concentration or type of extract included in the formulation should be mentioned., the encapsulating agent should be clearly specified. A new column with the type of assay should be mentioned. The main outcomes should be given in more details
L622: Replace ‘efficacies’ by ‘efficacious’
L639: Butea monosperma should be in italics. And the family name should be added.
L662: PLA should be introduced as abbreviation at L660.
L638-640: the authors should add some info about the gel composition.
L674: Define abbreviation SR. This section should be more detailed, with focus on the formulation. There is no info about that.
L675: Have the authors defined what is IIIM-160?
L734: ‘floral pharmacological active moieties’? What is that?
Round 2
Reviewer 1 Report
The paper has been improved therefore it is now suitable for publication
Reviewer 3 Report
Accept in current form